# In Vitro Model of the Human Blood–Brain Barrier to Explore HTLV-1 Immunopathogenesis

**DOI:** 10.3390/cimb47100818

**Published:** 2025-10-03

**Authors:** Ana Beatriz Guimarães, Lucas Bernardo-Menezes, Elisa Azevedo, Almerinda Agrelli, Poliana Silva, Marília Sena, Waldecir Araújo Júnior, George Diniz, Wyndly Daniel Gaião, Claudio Rodrigues, Marton Cavalcante, Lúcio Roberto Castellano, Joelma Souza, Paula Magalhães, Antonio Carlos Vallinoto, Clarice Morais

**Affiliations:** 1Laboratory of Virology and Experimental Therapy (LaViTE), Aggeu Magalhães Institute, Oswaldo Cruz Foundation (Fiocruz), Recife 50670-420, PE, Brazil; anabgiles@gmail.com (A.B.G.); elisaaalmeida@gmail.com (E.A.); almerindaapimentel@gmail.com (A.A.); poligomes.efl@gmail.com (P.S.); mariliasena97@hotmail.com (M.S.); waldeciroajunior@gmail.com (W.A.J.); 2Statistics and Geoprocessing Center, Aggeu Magalhães Institute, Oswaldo Cruz Foundation (Fiocruz), Recife 50670-420, PE, Brazil; george_tadeu@yahoo.com.br; 3Laboratory of Membrane Biophysics and Stem Cells-Dr. Oleg Krasilnikov, Department of Biophysics and Radiobiology, Federal University of Pernambuco (UFPE), Recife 50670-901, PE, Brazil; wyndly@hotmail.com (W.D.G.); claudio.rodrigues@ufpe.br (C.R.); 4Center for Technological Platforms, Aggeu Magalhães Institute, Oswaldo Cruz Foundation (Fiocruz), Recife 50670-420, PE, Brazil; martonc@fiocruz.br; 5Professional and Technological Center, Technical School of Health, Federal University of Paraíba (UFPB), João Pessoa 58051-900, PB, Brazil; lucio.castellano@academico.ufpb.br; 6Research Group in Biotechnology and Health (GePeMBiS), Federal University of Paraíba (UFPB), João Pessoa 58051-900, PB, Brazil; joelma.souza@academico.ufpb.br; 7HTLV Outpatient Clinic, Osvaldo Cruz University Hospital (HUOC), University of Pernambuco (UPE), Recife 50100-010, PE, Brazil; paulamagi@gmail.com; 8Laboratory of Virology, Institute of Biological Sciences, Federal University of Pará (UFPA), Belém 66075-110, PA, Brazil; vallinoto@ufpa.br

**Keywords:** endothelial permeability, human retrovirus, innate immunity, leukocyte influx, neuronal damage

## Abstract

Cellular components and inflammatory mediators involved in the transmigration of HTLV-1-infected cells across the blood–brain barrier (BBB) are not fully understood. This study proposes a BBB model to identify the immunological mechanisms associated with HTLV-1 pathogenesis. PBMCs from individuals with HTLV-1-associated Myelopathy/Tropical Spastic Paraparesis (HAM/TSP) (*n* = 4) or HTLV-1-infected individuals without HAM/TSP (*n* = 4) were isolated. An indirect cell co-culture was performed between human brain microvascular endothelial (hBMEC) cells and neuroblastoma (SH-SY5Y) cells. PBMCs from healthy individuals (*n* = 4) were used as a negative control, and MT-2 cells were used as a positive control. Supernatants and cells were collected to quantify inflammatory cytokines and assess cell death after 24, 48, and 72 h. Multiple comparisons were performed using the Kruskal–Wallis test, followed by Fisher’s LSD post hoc analysis. We observed that the production of cytokines IL-6, IL-8, IL-1β, TNF, IL-10, and IL-12p70, as well as the rate of neuronal death, was higher in co-cultures mimicking HAM/TSP carriers compared to HTLV-1-infected individuals without HAM/TSP and controls. Our results suggest that the HAM/TSP condition induces the release of IL-6, IL-8, IL-1β, TNF, IL-10, and IL-12p70, along with the infiltration of mononuclear cells, which may lead to neuronal death.

## 1. Introduction

Human T-lymphotropic virus (HTLV) is a positive-sense single-stranded RNA retrovirus of the Deltavirus genus, with four described types (-1, -2, -3, and -4) [1,2,3]. Among them, HTLV-1 is globally distributed, particularly in South America, the Caribbean, Africa, the Middle East, and parts of Asia and Australia, and is clinically significant due to its ability to establish chronic infection with persistent immunosuppression in some individuals. HTLV-1 is associated with severe inflammatory disorders, including Adult T-Cell Leukemia (ATL) and HTLV-1-Associated Myelopathy/Tropical Spastic Paraparesis (HAM/TSP) [4,5].

HAM/TSP affects a small proportion (0.3–3.8%) of HTLV-1 carriers and is characterized by chronic inflammation with focal lesions and mononuclear cell infiltration in the central nervous system (CNS). Clinically, it presents with progressive spastic weakness of the lower limbs, sensory disturbances, and autonomic dysfunctions including bladder and bowel symptoms [6,7,8]. The neuropathology includes spinal cord injury, demyelination, and neuronal degeneration, which are thought to result both from viral cytopathic effects and dysregulated immune responses [8].

The blood–brain barrier (BBB) serves as the critical interface between systemic circulation and the CNS, maintaining brain homeostasis by regulating molecular traffic and immune cell infiltration. BBB integrity depends on endothelial cells, pericytes, and astrocytes, and its disruption can facilitate the transmigration of HTLV-1-infected immune cells into the CNS [9,10]. In HTLV-1 infection, BBB dysfunction may permit infected cells and inflammatory mediators to enter the CNS, contributing to neurological damage and HAM/TSP pathogenesis [11,12,13].

HAM/TSP is characterized by enhanced migration of mononuclear cells and an inflammatory milieu dominated by Th1-type cytokines such as interferon-gamma (IFN-γ) and tumor necrosis factor (TNF), with reduced Th2 cytokines including interleukin (IL)-4 and IL-10 [8]. Elevated proviral load in infected CD4+ T cells correlates with increased risk of HAM/TSP development, underscoring the interplay between viral burden and immune dysregulation [14,15,16,17,18,19].

Despite these insights, the cellular and molecular mechanisms underlying HTLV-1-mediated BBB disruption and subsequent neuronal injury remain incompletely understood. Here, we propose an in vitro human BBB model to investigate the role of inflammatory cytokines in the immune-mediated pathogenesis of HAM/TSP, aiming to elucidate the mechanisms driving CNS tissue damage during HTLV-1 infection.

## 2. Materials and Methods

### 2.1. Study Population

The recruitment of HTLV-1-positive individuals was carried out at the HTLV Outpatient Clinic, Oswaldo Cruz University Hospital, University of Pernambuco, Recife, Pernambuco, Brazil. HTLV-1-negative individuals were recruited from the Aggeu Magalhães Institute (IAM) in Recife (Pernambuco, Brazil). All participants in the study were adults aged 18 years or older and had undergone testing to confirm their HTLV-1 infection status.

Healthy donors (*n* = 4) were randomly selected to serve as the negative control, ensuring unbiased representation and validity in the comparative analysis against HTLV-1-infected groups. Patients were randomly selected and grouped as seropositive for HTLV-1 without HAM/TSP (*n* = 4) or seropositive for HTLV-1 with HAM/TSP (*n* = 4). Within these groups, six participants were female (6F/2M) and averaged 51 (44–60) years old. Lower limb pain, back pain, arthralgia, and/or constipation were symptoms reported in HAM/TSP participants and HTLV participants without HAM/TSP. Loss of strength or movement in the lower limbs was an additional criterion for including participants as HAM/TSP, according to the Osame scale of motor incapacity [20].

### 2.2. Ethical Considerations

After reading and signing the Informed Consent Form, the participants voluntarily participated in the study. The study was conducted in accordance with the Declaration of Helsinki, and approved by the Ethics and Research Committee at Aggeu Magalhães Institute, Fiocruz—Pernambuco, on 13 May 2016 (Certificate of Presentation for Ethical Consideration: 50563115.9.0000.5190; Approval code: 1.508.891).

### 2.3. Sample Processing

Peripheral blood samples were collected by venipuncture in a 5 mL tube with EDTA using a vacuum system (BD Vacutainer^®^, Franklin Lakes, NJ, USA) to obtain the peripheral blood mononuclear cells (PBMCs). PBMCs were purified in a 50 mL Falcon using Ficoll Hypaque solution with a density of 1077 g/L (GE HealthCare^®^, Chicago, IL, USA) in a 3:4 blood ratio. In sequence, the content was centrifuged at 400× *g* for 10 min at 20 °C with the following collection of PBMC-formed rings.

In a 15 mL Falcon tube, phosphate-buffered saline (PBS) was added to the PBMCs for centrifugation at 400× *g* for 10 min at 20 °C. The supernatant was removed, and the cell pellet obtained was resuspended in 1 mL of RPMI 1640 medium (Sigma-Aldrich^®^, St. Louis, MO, USA), supplemented with 10% Fetal Bovine Serum (FBS, Gibco^®^, Waltham, MA, USA), as well as 1% Penicillin (100 UI/mL) and Streptomycin (100 µg/mL) (Pen/Strep, Sigma-Aldrich^®^, St. Louis, MO, USA).

### 2.4. Cell Culture

Human brain microvascular endothelial (hBMEC), neuroblastoma (SH-SY5Y), and HTLV-1 immortalized T (MT-2) cell lineages were all maintained at 37 °C with 5% CO_2_. Dulbecco’s Modified Eagle Medium with HAM/F12 1:1 (DMEM/F12, Gibco^®^, Waltham, MA, USA) was used to maintain SH-SY5Y. For the hBMEC cultures, medium 199 (M199, Gibco^®^, Waltham, MA, USA) was used. These media preparations were supplemented with 10% FBS (Gibco^®^, Waltham, MA, USA), 1% non-essential amino acids (NEAA, Sigma-Aldrich^®^, St. Louis, MO, USA), and 1% Pen/Strep (100 UI/mL:100 µg/mL, Sigma-Aldrich^®^, St. Louis, MO, USA). PBS (pH 7.2–7.4) and trypsin–EDTA solution (Sigma-Aldrich^®^, St. Louis, MO, USA) were also used during this cell culture’s maintenance. The MT-2 cell line was cultivated in RPMI 1640 (Sigma-Aldrich^®^, St. Louis, MO, USA) with 20% FBS, 2 mM L-glutamine (Sigma-Aldrich^®^, St. Louis, MO, USA), and 1% Pen/Strep (100 UI/mL:100 µg/mL, Sigma-Aldrich^®^, St. Louis, MO, USA).

### 2.5. Indirect Co-Cultivation Standard Procedure

A standard assay for SH-SY5Y cell concentrations was accomplished to define the ideal quantity of cells in indirect co-cultivation in 24-well flat-bottom plates (Kasvi^®^, Shenzhen, China), with a surface area of 1.9 cm^2^ per well, as follows: 100,000 cells/well, 150,000 cells/well, and 200,000 cells/well. The hBMEC line was added to inserts with a 3 μm semi-permeable porous membrane and culture surface of 0.336 cm^2^ (Greiner^®^, Pleidelsheim, Germany), in cell concentrations of 10,000 cells/well, 30,000 cells/well, 50,000 cells/well, 80,000 cells/well, and 100,000 cells/well. To perform the standard assay, incubation periods were up to 6 days and were accomplished in quadruplicate.

### 2.6. Mimetic-Human BBB

An in vitro BBB model was mimicked using inserts to build an apical compartment in each well of the 24-well plate that acted as a basolateral compartment [13]. The hBMEC line (M199, 1% Pen/Strep, 1% NEAA, and 2% FBS) was seeded in the inserts, and SH-SY5Y cells (DMEM/F12, 1% Pen/Strep, 1% NEAA, and 10% FBS) were seeded in a 24-well plate. Both cell lines were maintained under the previously detailed conditions. After 24 h, the inserts were carefully added to each well of the plate. In the apical compartment, 1 × 10^5^ cells/well of PBMCs or 1 × 10^5^ cells/well of the MT-2 line (RPMI 1640, 1% Pen/Strep, and 10% FBS) were added as stimuli. MT-2 cells were seeded in the inserts in quadruplicate. The analyses were performed after 24, 48, and 72 h of incubation (Figure 1).

The stimulus conditions were performed in four groups as follows: [a] negative control (PBMCs from healthy individuals); [b] positive control (MT-2 cells, which are immortalized cells infected by HTLV-1); [c] PBMCs from individuals seropositive for HTLV-1 without HAM/TSP; and [d] PBMCs from individuals seropositive for HTLV-1 with HAM/TSP. At the conclusion of each incubation period within the BBB co-culture model, both the supernatant and corresponding cellular components were collected for subsequent analysis of inflammatory cytokine levels and assessment of cell death.

### 2.7. Assessing Barrier Integrity

Transendothelial electrical resistance (TEER) was measured using a patch-clamp electrophysiology system (Axopatch 200B; Axon Instruments/Molecular Devices Corp., Union City, CA, USA) controlled by Clampex software version 10.5.1.0 (Membrane Test module). The electrical signals were digitized and transmitted to a computer via the Digidata 1550A interface (Axon Instruments). The hBMEC line (M199, 1% Pen/Strep, 1% NEAA, and 2% FBS) was seeded in the inserts (0.8 × 10^5^ cells), and SH-SY5Y cells (DMEM/F12, 1% Pen/Strep, 1% NEAA, and 10% FBS) were seeded in a 24-well plate (1.5 × 10^5^ cells). After 24 h, the inserts containing hBMECs were carefully transferred into the wells containing SH-SY5Y cells. Hand-held electrodes were inserted into the apical compartment (hBMEC-seeded insert) and basolateral compartment (SH-SY5Y-seeded 24-well plate). A −40 mV electrical pulse was applied across the cell monolayers, and recordings were acquired for 50 microseconds in triplicate. TEER values ± standard error of mean were calculated and expressed as ohms per square centimeter (Ω·cm^2^).

### 2.8. Quantification of Inflammatory Cytokines

Following the manufacturer’s instructions, cellular supernatant from indirect co-culture was used to evaluate the cytokine levels using the BD CBA Human Inflammatory Cytokines Kit (BD Biosciences, Franklin Lakes, NJ, USA). IL-12p70, TNF, IL-10, IL-6, IL-1β, and IL-8 were measured by FACS Calibur Flow Cytometer (BD Biosciences^®^, Franklin Lakes, NJ, USA), and their results were analyzed through FCAP Array v3 software (SoftFlow Inc., New Brighton, MN, USA).

### 2.9. Cell Death Assay

To evaluate the cell death profile in SH-SY5Y cells from the BBB mimic model, FITC Annexin V Apoptosis Detection Kit (BD Pharmingen^®^, Franklin Lakes, NJ, USA) was used following the manufacturer’s instructions. Analyses were executed by FACS Calibur Flow Cytometry (BD Biosciences^®^, Franklin Lakes, NJ, USA), and results were generated using FCAP Array v3 software (SoftFlow Inc., New Brighton, MN, USA).

### 2.10. Statistical Analysis

We performed a statistical analysis of the obtained data using R software version 4.5 (R Core Team, Vienna, Austria), setting our significance threshold at *p* < 0.05. To evaluate multiple comparisons, we used the Kruskal–Wallis test, followed by Fisher’s LSD post hoc analysis with Bonferroni correction. To clearly present our findings, we created data visualizations that highlight the means and medians.

## 3. Results

### 3.1. Innate Immunity-Mediated Inflammatory Response Regulates the BBB Cells

The results indicate that, between 24 and 72 h, only the concentration of 0.8 × 10^5^ cells/well for hBMEC (apical compartment) and 1.5 × 10^5^ cells/well for SH-SY5Y maintained a confluence between 70% and 90% (basolateral compartment).

Our previously established human BBB model demonstrated a TEER value of 4714.08 ± 44.80 Ω·cm^2^. Cellular supernatant from the BBB mimic model stimulated with MT-2, HTLV-1-infected PBMCs, or HTLV-1-uninfected PBMCs was utilized to measure the following inflammatory cytokines: IL-12p70, TNF, IL-10, IL-1β, IL-6, and IL-8 (Table 1).

The statistical data regarding the comparisons between the groups and the analyzed cytokines are available Appendix A. We observed that IL-6, IL-8, IL-1β, TNF, IL-10, and IL-12p70 increased for both cell compartments. IL-6, IL-1β, and TNF are cytokines mainly involved in the acute phase of inflammation [11,14]. Therefore, these pro-inflammatory cytokines may significantly affect the initial immune response to viral infection alongside IL-12 [8]. In addition, IL-8 may lead to effective action on the intracellular antigen, but in contrast, it may facilitate viral proliferation through the cell-to-cell virus mechanism of infection [21]. Finally, IL-10 acts as an anti-inflammatory cytokine by restricting the infiltration of mononuclear cells across the BBB induced by HTLV-1 infection, thereby helping to limit neuroinflammation [8].

#### 3.1.1. Pro- and Anti-Inflammatory Cytokines Modulate the In Vitro BBB Endothelium by Cellular Stimuli

In the hBMEC line from the BBB model, stimulation with PBMCs from HTLV-1-infected individuals (both HTLV+ HAM/TSP+ and HTLV+ HAM/TSP− groups) induced significantly higher cytokine levels compared to positive (MT-2 cells) and negative (healthy PBMCs) controls at all time points (24, 48, 72 h).

IL-6 was markedly elevated in both of the groups of infected individuals, with the HTLV+ HAM/TSP+ group showing the highest and sustained production (*p* = 0.0042) and the HTLV+ HAM/TSP− group exhibiting a peak at 48 h (*p* = 0.0450). IL-8 increased over time, peaking at 48 h, with both of the groups of infected individuals significantly above controls (*p* < 0.005). IL-1β levels were also significantly higher in the groups of HTLV-1-infected individuals across the entire time (*p* ≤ 0.0004), peaking at 48 h and remaining elevated at 72 h, indicating sustained endothelial activation.

TNF secretion showed a temporal distinction: the HTLV+ HAM/TSP+ group peaked early at 24 h, whereas the HTLV+ HAM/TSP− group peaked moderately at 48 h. Controls exhibited consistently low or intermediate levels. Both infected groups had significantly elevated TNF compared to controls (*p* < 0.005), reflecting a strong pro-inflammatory response.

IL-10, an anti-inflammatory cytokine, was significantly elevated in the HTLV+ HAM/TSP+ group with a peak at 24 h, declining thereafter, while the HTLV+ HAM/TSP− group maintained low stable levels. Controls showed minimal or decreasing IL-10 (*p* ≤ 0.0001), suggesting anti-inflammatory feedback in HAM/TSP conditions.

Finally, IL-12p70 was significantly increased in both infected groups versus controls at all time points (*p* ≤ 0.0001). The HTLV+ HAM/TSP− group peaked at 24 h with a gradual decline, while the HTLV+ HAM/TSP+ group exhibited a sustained peak at 48–72 h. Controls had negligible IL-12p70 production.

Overall, these data demonstrate a robust, sustained endothelial inflammatory response triggered by PBMCs from HTLV-1-infected individuals, especially those with HAM/TSP. The simultaneous increase in IL-10 suggests a regulatory mechanism to counterbalance inflammation. These cytokine patterns support the concept of complex immune modulation by HTLV-1 in the BBB microenvironment, potentially contributing to the neuroinflammatory pathology observed in HTLV-1-associated neurological disease.

#### 3.1.2. Innate Immune Response from BBB Model Contributes to Neuroinflammatory Progression in HAM/TSP

We found that significant IL-6 levels were detected in supernatants from SH-SY5Y cells within the BBB mimic system stimulated with PBMCs from HTLV-1-seropositive individuals with HAM/TSP. IL-6 concentrations increased over time (24 to 72 h), peaking in the HTLV+ HAM/TSP+ group, which showed significantly higher levels than the HTLV+ HAM/TSP− group (*p* ≤ 0.01) and both positive (MT-2 cells) and negative controls (*p* ≤ 0.0001). The HTLV+ HAM/TSP− group exhibited intermediate IL-6 levels, suggesting a gradient of immune activation correlated with clinical status. Control groups displayed minimal IL-6 production, supporting the specificity of the inflammatory response to HAM/TSP.

IL-8 secretion in SH-SY5Y supernatants also varied by clinical status and incubation time. Both HTLV-1-infected groups showed elevated IL-8 levels, peaking at 72 h, and with significantly higher concentrations in the HTLV+ HAM/TSP+ group than controls at all time points (*p* ≤ 0.0001). Control groups maintained low and stable IL-8 levels consistent with basal inflammatory activity. IL-1β was markedly elevated in the cells stimulated by PBMCs from HTLV-1-infected individuals, particularly in the HAM/TSP+ group at 48 and 72 h (*p* ≤ 0.0001), indicating sustained pro-inflammatory activation.

TNF levels also were significantly increased in neuronal co-cultures stimulated with PBMCs from HTLV-1-infected individuals compared to controls across all time points. The HTLV+ HAM/TSP+ group showed a pronounced TNF peak at 24 and 48 h, which declined by 72 h (*p*-values ranging from 0.0002 to 0.0209). Controls exhibited low and stable TNF levels throughout.

IL-10 secretion was significantly elevated in both HTLV+ HAM/TSP+ and HTLV+ HAM/TSP− groups versus controls (*p* < 0.0001), with the highest levels in the HAM/TSP+ group at 48 h, reflecting an active regulatory response to balance inflammation. Controls produced negligible IL-10. Whereas IL-12p70 was undetectable in controls but measurable in neuronal co-cultures when stimulated with PBMCs from both HTLV-1 groups at all times, the HTLV+ HAM/TSP+ group showed a prominent IL-12p70 peak at 48 h (significant compared to controls, *p* ≤ 0.0409).

Overall, the sustained and increasing levels of IL-6, IL-8, IL-1β, TNF, and IL-12p70 observed in the HTLV+ HAM/TSP+ group reveal a strong pro-inflammatory environment within the basolateral compartment of the BBB model. At the same time, the rise in IL-10 points to an attempt by the system to counterbalance this inflammation, although this anti-inflammatory response appears insufficient to fully resolve the immune activation. The low cytokine levels seen in the control groups highlight that these effects are specific to HTLV-1 infection and the HAM/TSP condition. Together, these results highlight a complex cytokine network mediating communication between endothelial and immortalized neuronal cells in the context of HTLV-1-related neuroinflammation.

### 3.2. HTLV-1-Infected PBMCs Trigger Neuronal Apoptosis in a BBB Mimic System

Assessment of cell death in the SH-SY5Y cells from the BBB model revealed that the group exposed to PBMCs from individuals seropositive for HTLV-1 with HAM/TSP exhibited the highest rates of neuronal apoptosis between 24 h and 72 h of incubation (Table 2). Increased cell death levels were also noted in those exposed to PBMCs from individuals seropositive for HTLV-1, with or without HAM/TSP, and both positive and negative controls (Figure 2).

In the negative control group, the rate of neuronal apoptosis increased between 24 and 48 h (*p* = 0.0034) and between 24 and 72 h (*p* = 0.0039). At 24 h of incubation, this group exhibited lower apoptosis rates compared to both the positive control at 48 and 72 h (*p* = 0.0013 and *p* = 0.0129, respectively) and the groups stimulated with PBMCs from seropositive individuals for HTLV-1, with or without HAM/TSP, at the same time points (*p* < 0.0005).

Within the positive control group, apoptosis at 24 h was lower than at 48 and 72 h (*p* = 0.0051 and *p* = 0.0472). Additionally, apoptosis at 24 h was lower than that observed at 72 h in the groups stimulated by PBMCs from individuals with and without HAM/TSP (*p* = 0.0011 and *p* = 0.0128, respectively). Conversely, at 72 h, the apoptosis rate in the positive control was higher than in the HTLV+ HAM/TSP+ and HTLV+ HAM/TSP− groups at 24 h (*p* = 0.0069 and *p* = 0.0078, respectively).

In the group stimulated by PBMCs from seropositive individuals for HTLV-1 without HAM/TSP, neuronal apoptosis increased from 24 to 72 h (*p* = 0.0026), with apoptosis at 72 h exceeding that in the HTLV+ HAM/TSP+ group at 24 h (*p* = 0.0002). In the HTLV+ HAM/TSP+ group, apoptosis also showed an increase between 24 and 48 h (*p* = 0.05) and from 24 to 72 h (*p* = 0.0001).

In summary, SH-SY5Y cells incubated with PBMCs from individuals who are seropositive for HTLV-1 with HAM/TSP exhibited the highest apoptosis rates, particularly at 24 and 72 h. In contrast, the HTLV+ HAM/TSP− and control groups showed comparatively lower levels of neuronal cell death. These findings suggest that PBMCs from patients who are seropositive for HTLV-1 with HAM/TSP possess a heightened pro-inflammatory potential and an increased capacity to induce progressive neuronal apoptosis, corroborating previous studies [12,13].

## 4. Discussion

### 4.1. Cytokines and BBB Permeability

HAM/TSP is a disabling neurological condition, with about 30% of affected individuals becoming paraplegic and bedridden within ten years of symptom onset. Beyond cellular and molecular damage to CNS cells, the host’s innate immune response plays a crucial role in disease development. Thus, inflammatory mediators, cytokines, and BBB permeability are key to understanding its immunopathology [4,5,6,7,8,9,10].

Our results indicate that HTLV-1 triggers an innate immune response primarily mediated by cytokines IL-6, IL-8, IL-1β, TNF, IL-10, and IL-12p70. Stimulation of immortalized BBB cell lines with PBMCs from HTLV-1-seropositive individuals with HAM/TSP consistently induced higher inflammatory cytokine levels. The co-culture model using hBMEC and SH-SY5Y cells appears to have increased permeability when exposed to HTLV-1-infected cells. Moreover, our indirect co-culture system demonstrated TEER values higher than those previously reported for immortalized human brain endothelial cells, reflecting robust barrier integrity [22].

Consistent with previous findings on HTLV-1 persistent lymphocytes, our results show that inflammatory cytokines (including IL-6, IL-8, IL-1β, TNF, IL-10, and IL-12p70) are overproduced in both hBMEC and SH-SY5Y cells upon HTLV-1 infection [13]. Afonso et al. also investigated BBB disruption using an in vitro model and found that endothelial cells can be infected by HTLV-1, causing phenotypic changes, suggesting that infected lymphocytes may cross the barrier and enter the CNS [13].

HTLV-1-infected cells modulate other target cells, disseminating viral proteins through extracellular vesicles during cell-to-cell contact, thereby contributing to increased cellular injury and inflammatory damage [23]. In an ultrastructural analysis using microvascular endothelial cells and the MT-2 cell line, it was suggested that HTLV-1 particles may cross the BBB endothelium by a transcytosis pathway. Cellular contact between the two cell lineages can lead to accelerated budding of virions from lymphocytes, their internalization in endothelial vesicles, and the release of additional viral particles [24].

Besides serving as viral reservoirs, HTLV-1-infected CD4+ and CD8+ T cells can produce both pro- and anti-inflammatory cytokines during infection progression [25]. In our study, IL-12p70 levels varied across experimental groups. Some reports suggest higher IL-12 gene expression in PBMCs from HTLV-1-seropositive individuals without HAM/TSP compared to those with HAM/TSP [26]. While IL-12 promotes T cell proliferation and supports the release of other antiviral cytokines [27], it may also play a protective role in CNS neuroinflammation [28]. Unlike many other viral infections where IL-12 rises early, we did not observe a significant IL-12 increase, highlighting a possible difference in immune response dynamics in HTLV-1 infection [29].

TNF is sometimes found at reduced levels in HTLV-1-seropositive individuals, which may aid viral persistence alongside IL-6 [30]. IL-6 itself promotes immune evasion and the proliferation of infected cells [31,32], while IL-8, released by infected PBMCs, contributes to inflammation by increasing interactions between neutrophils and T lymphocytes, potentially leading to cell death [25]. In our in vitro BBB model, IL-6, IL-8, IL-1β, and TNF emerged as key mediators of cellular damage during HTLV-1 immunopathogenesis. Based on cytokine kinetics, these molecules, possibly together with IL-12p70 and IL-10, could serve as biomarkers of the acute inflammatory response to HTLV-1 infection in the BBB microenvironment over time.

IL-6, a key acute-phase pro-inflammatory cytokine, was significantly elevated in both cell lines stimulated with PBMCs from HTLV-1-seropositive patients with HAM/TSP. This supports previous reports suggesting a dual role for IL-6 in viral infections: it contributes to antiviral defense but can also promote viral replication. Additionally, IL-6 is known to amplify inflammation and neuroinflammation in diseases such as multiple sclerosis and HIV-associated neurocognitive disorders [33,34,35].

### 4.2. Mechanisms of Neuronal Damage

In addition to HAM/TSP, ATL is linked to dysregulation of cytokines like IL-10. Lower IL-10 levels have been associated with neurological complications from HTLV-1 infection [31]. In our study, however, IL-10 levels were low, which we suggest reflects its role as an immunosuppressive and neuroprotective cytokine.

One of the key findings in our study was the higher release of IL-10 in both cell lines stimulated with PBMCs from HTLV-1-seropositive individuals with HAM/TSP. IL-10 is recognized for its role in limiting excessive immune responses due to its anti-inflammatory effects [36]. Our results align with previous studies showing increased IL-10 production during viral infections [37,38,39,40].

A cross-sectional study found that CD4+ T cell counts in HIV-1-infected individuals correlate with CNS injury [41]. Among 101 HIV-1 patients, neuronal damage was linked to the infiltration of neurotoxic molecules and proviral particles into the CNS [42]. Similarly, inflammatory mediators may contribute to spinal cord and brain changes in HAM/TSP patients, where early lesions are dominated by CD4+ T cells but shift to CD8+ T cell predominance as the disease progresses [8]. Despite this, the exact inflammatory mechanisms behind neuronal damage in HTLV-1 infection are still not fully understood.

Here, we also show a clear association between increased neuronal death in vitro and the HAM/TSP condition. In contrast, SH-SY5Y cells stimulated with MT-2 cells or healthy PBMCs exhibited lower apoptosis. This supports the link between viral infection severity, clinical condition, and neuronal damage in our BBB model.

As noted previously, MT-2 cells and PBMCs differ in cellular composition, which may explain why MT-2 cells did not cause greater neural injury than expected. Additionally, cytokine responses can vary in primary cells due to their genetic and phenotypic differences from immortalized lines.

Our data suggest that the inflammatory response triggered by systemic HTLV-1 infection, combined with interactions between infected cells and BBB cells, is enough to increase cytokine levels, and potentially other biomarkers, altering the immune environment in the in vitro BBB. We believe this study provides initial insight into a BBB model for HTLV-1 research, an area currently lacking in the literature.

Maldonado and colleagues [32] visualized neuronal damage in SH-SY5Y cells through soluble viral proteins in cerebrospinal fluid from HAM/TSP carriers or by MT2 supernatant. In this context, it was observed that HTLV-1-infected cells may secrete the Tax protein, which acts as a stimulus for neurite retraction in vitro [43,44]. HTLV-1 preferentially can infect neuronal cells in human stem cell-derived polycultures, suggesting a local immune response, and thus, a neuroinfection could be possible [43,44]. To date, studies on neuronal cell death using HTLV-1-infected cells in an in vitro BBB model have not been described.

Several studies have examined cell death caused by HIV infection using the human SH-SY5Y neuronal cell line [45,46,47,48], but none have focused on the human BBB. This highlights the potential of our work to extend to studies involving other viruses. SH-SY5Y cells are a widely used, undifferentiated human-derived line known for modeling neuronal interactions. Their use supports the physiological integrity of co-cultures in human BBB models and advances understanding of neurological mechanisms in CNS diseases [47,48,49].

### 4.3. Comparison with Previous In Vitro Studies

Studies using human cerebral microvascular endothelial cells have shown that TNF and IL-1β work together to increase BBB permeability by stimulating leukocyte adhesion molecules like VCAM-1 and ICAM-1 [24,50]. TNF also contributes to tissue degradation in the CNS of HAM/TSP patients by regulating matrix metalloproteinases [28,31]. Alongside TNF, elevated IL-1β appears to play a key role in neuronal damage, influenced by the host’s genetic background and the clinical progression of HTLV-1 infection [33]. Our findings align with previous reports of high IL-1β production by PBMCs from HAM/TSP patients in the BBB model.

Both cell lines showed a marked increase in IL-8 in response to HTLV-1 infection, highlighting its role in immune cell recruitment. Previous studies have demonstrated that IL-8 attracts inflammatory cells to the CNS during viral infections like HIV, intensifying inflammation and contributing to neuronal damage [21,49]. Our findings may suggest that IL-8 similarly drives the immune response to HTLV-1, promoting inflammation and possibly facilitating viral spread within the CNS.

Our study showed significant neuronal loss in SH-SY5Y cells under HTLV-1-infected conditions, consistent with previous reports linking viral infections like HTLV-1 to direct neuronal death and neurodegeneration [51,52,53,54,55,56,57]. TNF is a key modulator of BBB permeability, contributing to neuronal damage [52,58]. Interactions between brain endothelial cells and HTLV-1-infected lymphocytes may elevate TNF levels at the BBB [24], with TNF increasing barrier permeability through the RhoA/Rho-kinase pathway within hours up to 24 h after viral exposure [59]. Additionally, dendritic cells stimulated with the HTLV-1 Tax protein show increased IL-6, IL-8, IL-1β, and TNF, supporting their role in neuronal damage during HAM/TSP [60].

While our in vitro BBB mimic model provides important insights into HTLV-1 immunopathogenesis, it simplifies the human BBB’s complex biology by including only endothelial and neuroblastoma cells. Cellular components such as astrocytes, pericytes, microglia, and the extracellular matrix are absent, as well as physiological factors like blood flow and shear stress. This BBB model may not fully reflect in vivo conditions or the complex cellular interactions in the neurovascular unit. Therefore, our findings should be interpreted with caution and validated in more physiologically relevant co-culture or animal models.

Additionally, it is essential to expand the methods used to assess barrier permeability by incorporating complementary functional assays, allowing for selective identification of the molecules and cells that cross the BBB. Evaluating a broader range of cytokines and inflammatory mediators over extended time periods will also help to more clearly elucidate the molecular mechanisms underlying neuroinflammatory injury and neuronal death associated with HTLV-1 infection.

### 4.4. Implications for Future Research

Our model demonstrated a marked elevation of pro-inflammatory cytokines—IL-6, IL-8, IL-1β, and TNF—along with increased neuronal apoptosis, mirroring the neuroinflammatory profile and neurological deterioration observed in HAM/TSP patients. These cytokines may serve as crucial biomarkers in peripheral blood or cerebrospinal fluid for monitoring disease progression and evaluating treatment efficacy related to BBB integrity and neuronal homeostasis.

This translational study showed that the inflammatory response of innate immunity and neuronal damage act as potent molecular and cellular mechanisms in HTLV-1 infection. Moreover, those reports enhance the scientific community’s understanding of the influx of human retroviruses or neurotropic viruses into the CNS via the BBB. We believe that due to the scarcity of data in the literature, our preliminary results might encourage future studies to elaborate new experimental designs addressing the BBB mimic model in HTLV-1 infection.

One important limitation of this study is the relatively small sample size. To enhance the translational impact and to validate the mechanisms suggested by our results, future research must involve larger, more diverse participants. In addition, our study contributes to a better understanding of HTLV-1-induced immunopathogenesis at the BBB interface and identifies promising biomarkers and therapeutic targets. Thus, a comprehensive validation through studies with larger cohorts and more sophisticated in vitro and in vivo models is necessary to confirm these early findings and advance toward clinical application.

Due to the small sample size, a non-parametric strategy was selected, with a Bonferroni correction applied to ensure the accuracy of the findings. It is essential to acknowledge that a sample size of four per group may affect the robustness of the results, as donor-specific genetic variability could introduce bias, influencing the findings. Furthermore, we highlight that the BBB model is an in vitro model and, therefore, limited. It should be used with caution in preliminary studies to guide more robust in vivo and clinical research. Addressing these limitations in future studies is crucial to enhancing the model’s relevance and translational value.

It is essential to develop in vivo systems, particularly relevant animal models, that would examine the effect of HTLV-1 on BBB integrity and neuroinflammation to explore the temporal dynamics of cytokine-mediated neuronal damage. We also need clinical validation to confirm the role of these cytokines as true disease biomarkers, ensuring findings are not limited to artifacts of the in vitro BBB model. Further clinical studies should explore whether the inflammatory cytokines we identified in our study (IL-6, IL-8, IL-1β, TNF, IL-10, and IL-12p70) could potentially serve as biomarkers for disease progression and response to therapy in HAM/TSP patients.

Intervention with therapeutics should concurrently be tested in these in vivo models to evaluate whether blockade or inhibition of pro-inflammatory mediators, notably IL-6, IL-8, IL-1β, or TNF, can lead to attenuated BBB disruption and neuronal damage. By integrating in vivo validation and therapeutic trials, these studies may provide important insights into both disease mechanisms and therapeutic approaches for HTLV-1-associated neurological diseases.

## 5. Conclusions

The data presented here offer important new knowledge regarding both the inflammatory processes associated with HTLV-1 infection and the effect of HTLV-1 on the BBB mimic model. Our experiments showed that from 24 h to 72 h, there were high levels of cytokines IL-6, IL-8, IL-1β, TNF, IL-10, and IL-12p70, especially in cells stimulated by the PBMCs from individuals seropositive for HTLV-1 with HAM/TSP, indicating an important role for these cytokines in the early response of the immune system to HTLV-1 infection. Cell death assays showed that, considering the three time points analyzed, PBMCs from seropositive HTLV-1 individuals with HAM/TSP killed more neuronal cells than the PBMCs from seropositive HTLV-1 individuals without HAM/TSP, the positive control (MT-2 cells), and the negative control (PBMCs from healthy individuals). Therefore, the innate immune response seems to play a crucial role in cell death in HTLV-1 infection and may show potential cytokines as biomarkers for monitoring disease and treatment response. We also need in vitro, in vivo, and clinical studies to validate and confirm the role of these cytokines in HTLV-1-associated neurological diseases. Thus, it is crucial to test therapeutic interventions targeting these inflammatory pathways to prevent BBB disruption and neuronal damage in HAM/TSP patients.

## Figures and Tables

**Figure 1 cimb-47-00818-f001:**
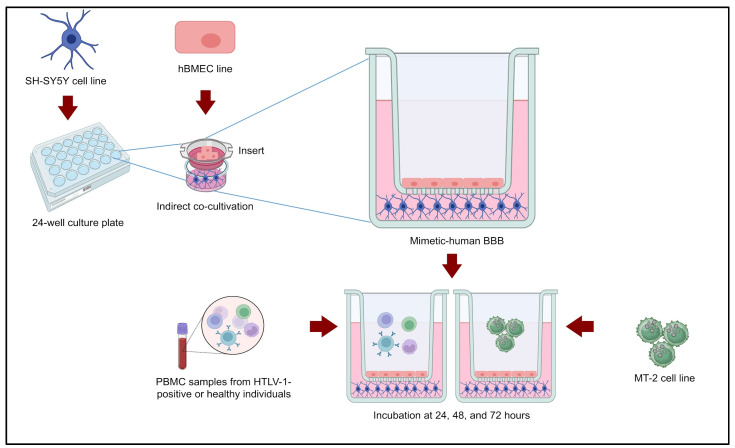
In vitro mimetic model of the human blood–brain barrier. A co-culture model including a neuronal cell line (SH-SY5Y) and an endothelial cell line (hBMEC) was used. The different forms of stimulation for HTLV-1 infection/immune response models were PBMCs from healthy donors, PBMCs from seropositive HTLV-1 donors (with and without HAM/TSP), and an HTLV-1 positive T lymphocyte cell line (MT-2).

**Figure 2 cimb-47-00818-f002:**
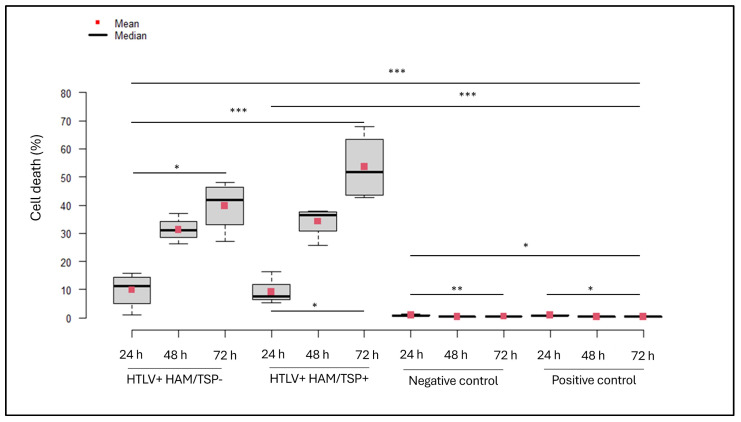
Neuronal apoptosis was evaluated in an in vitro BBB co-culture model consisting of SH-SY5Y neuronal cells and hBMEC endothelial cells. The system was stimulated with PBMCs from healthy donors (negative control), PBMCs from HTLV-1-seropositive individuals with HAM/TSP (HTLV+ HAM/TSP+), PBMCs from HTLV-1-seropositive individuals without HAM/TSP (HTLV+ HAM/TSP−), and the HTLV-1-positive T-cell line MT-2 (positive control). Statistical comparisons were performed using the Kruskal–Wallis test followed by Fisher’s LSD post hoc with Bonferroni correction (* *p* < 0.05; ** *p* < 0.005; *** *p* < 0.0005).

**Table 1 cimb-47-00818-t001:** Cytokine levels were measured in cell lines derived from the blood–brain barrier (BBB) model after co-culture stimulation with the following: (a) PBMCs from healthy donors (negative control); (b) MT-2 cells (immortalized HTLV-1-infected cells, positive control); (c) PBMCs from HTLV-1-seropositive individuals without HAM/TSP; (d) PBMCs from HTLV-1-seropositive individuals with HAM/TSP. Measurements were performed at 24, 48, and 72 h of incubation.

Cell Compartment	Time	Cytokines	Mean of Quantification (pg/µL)			
			Negative Control (*n* = 4)	Positive Control (MT-2 Cells)	HTLV+ HAM/TSP− (*n* = 4)	HTLV+ HAM/TSP+ (*n* = 4)
hBMEC line						
	24 h	IL-12p70	0	0.06	17.25	7.60
		TNF	721.45	57.76	88.61	2250.08
		IL-10	13.41	0	9.45	63.75
		IL-6	12,350.11	11,848.32	2,622,369.02	11,281,468.06
		IL-1β	996.48	19.82	6292.02	14,425.55
		IL-8	23,773.84	12,396.19	28,920.41	34,775.30
	48 h	IL-12p70	0	0.17	6.90	10.14
		TNF	135.08	22.10	242.58	782.57
		IL-10	3.49	0	10.34	30.89
		IL-6	12,290.31	11,906.03	10,167,982.96	10,410,740.41
		IL-1β	990.17	19.82	9881.23	19,174.38
		IL-8	14,711.46	15,370.40	48,510.48	50,203.04
	72 h	IL-12p70	0	1.25	5.21	9.50
		TNF	32.78	12.01	79.76	209.30
		IL-10	0.28	0	6.81	17.78
		IL-6	12,291.21	11,539.45	7,249,746.83	10,596,709.93
		IL-1β	738.29	40.80	6757.40	14,618.64
		IL-8	22,237.97	13,970.70	49,066.70	46,673.98
SH-SY5Y cells						
	24 h	IL-12p70	0	0	5.40	4.72
		TNF	162.71	15.62	131.70	117.86
		IL-10	2.74	0	6.94	10.71
		IL-6	12,504.31	8759.12	420,576.35	7,827,617.52
		IL-1β	246.24	0	4499.44	2674.93
		IL-8	22,758.84	8944.92	25,013.93	31,979.71
	48 h	IL-12p70	0	0	4.86	18.25
		TNF	45.64	10.44	65.24	95.71
		IL-10	1.81	0	6.99	13.07
		IL-6	9190.90	10,571.36	3,400,817.07	10,602,510.19
		IL-1β	460.01	9.98	3825.55	7974.24
		IL-8	8944.92	12,485.92	42,245.11	45,132.37
	72 h	IL-12p70	0	8.02	5.48	6.86
		TNF	20.45	3.61	50.17	58.77
		IL-10	0.74	0	6.26	12.00
		IL-6	13,062.54	10,760.72	7,297,915.70	12,406,516.19
		IL-1β	617.44	14.88	5135.01	8782.95
		IL-8	24,822.20	12,165.40	46,664.63	49,356.05

**Table 2 cimb-47-00818-t002:** Cell death in SH-SY5Y cells from the BBB model was assessed after 24, 48, and 72 h of stimulation with: PBMCs from healthy donors (negative control); MT-2 cells (positive control); PBMCs from HTLV-1-seropositive individuals without HAM/TSP (HTLV+ HAM/TSP−); and PBMCs from HTLV-1-seropositive individuals with HAM/TSP (HTLV+ HAM/TSP+).

Comparisons	Mean Difference in Cell Death (%)	*p*-Value
HTLV+ HAM/TSP− 24 h and HTLV+ HAM- 72 h	−13.5	0.0017
HTLV+ HAM/TSP− 24 h and HTLV+ HAM+ 72 h	−17.0	<0.0005
HTLV+ HAM/TSP− 24 h and Positive control 48 h	21.0	<0.0005
HTLV+ HAM/TSP− 24 h and Positive control 72 h	18.88	<0.0005
HTLV+ HAM/TSP− 24 h and Negative control 48 h	20.13	<0.0005
HTLV+ HAM/TSP− 24 h and Negative control 72 h	20.0	<0.0005
HTLV+ HAM/TSP− 48 h and Positive control 24 h	16.5	0.0001
HTLV+ HAM/TSP− 48 h and Positive control 48 h	29.0	<0.0005
HTLV+ HAM/TSP− 48 h and Positive control 72 h	26.88	<0.0005
HTLV+ HAM/TSP− 48 h and Negative control 24 h	15.25	0.0003
HTLV+ HAM/TSP− 48 h and Negative control 48 h	28.13	<0.0005
HTLV+ HAM/TSP− 48 h and Negative control 72 h	28.0	<0.0005
HTLV+ HAM/TSP− 72 h and HTLV+ HAM+ 24 h	13.75	0.0013
HTLV+ HAM/TSP− 72 h and Positive control 24 h	22.0	<0.0005
HTLV+ HAM/TSP− 72 h and Positive control 48 h	34.5	<0.0005
HTLV+ HAM/TSP− 72 h and Positive control 72 h	32.38	<0.0005
HTLV+ HAM/TSP− 72 h and Negative control 24 h	20.75	<0.0005
HTLV+ HAM/TSP− 72 h and Negative control 48 h	33.63	<0.0005
HTLV+ HAM/TSP− 72 h and Negative control 72 h	33.5	<0.0005
HTLV+ HAM/TSP+ 24 h and HTLV+ HAM+ 72 h	−17.25	<0.0005
HTLV+ HAM/TSP+ 24 h and Positive control 48 h	20.75	<0.0005
HTLV+ HAM/TSP+ 24 h and Positive control 72 h	18.63	<0.0005
HTLV+ HAM/TSP+ 24 h and Negative control 48 h	19.88	<0.0005
HTLV+ HAM/TSP+ 24 h and Negative control 72 h	19.75	<0.0005
HTLV+ HAM/TSP+ 48 h and Positive control 24 h	18.0	<0.0005
HTLV+ HAM/TSP+ 48 h and Positive control 48 h	30.5	<0.0005
HTLV+ HAM/TSP+ 48 h and Positive control 72 h	28.38	<0.0005
HTLV+ HAM/TSP+ 48 h and Negative control 24 h	16.75	<0.0005
HTLV+ HAM/TSP+ 48 h and Negative control 48 h	29.63	<0.0005
HTLV+ HAM/TSP+ 48 h and Negative control 72 h	29.5	<0.0005
HTLV+ HAM/TSP+ 72 h and Positive control 24 h	25.5	<0.0005
HTLV+ HAM/TSP+ 72 h and Positive control 48 h	38.0	<0.0005
HTLV+ HAM/TSP+ 72 h and Positive control 72 h	35.88	<0.0005
HTLV+ HAM/TSP+ 72 h and Negative control 24 h	24.25	<0.0005
HTLV+ HAM/TSP+ 72 h and Negative control 48 h	37.13	<0.0005
HTLV+ HAM/TSP+ 72 h and Negative control 72 h	37.0	<0.0005
Positive control 24 h and Positive control 48 h	12.5	0.0051
Positive control 24 h and Positive control 72 h	10.38	0.0472
Positive control 24 h and Negative control 48 h	11.63	0.0129
Positive control 24 h and Negative control 72 h	11.5	0.0148
Positive control 48 h and Negative control 24 h	−13.75	0.0013
Positive control 72 h and Negative control 24 h	−11.63	0.0129
Negative control 24 h and Negative control 48 h	12.88	0.0034
Negative control 24 h and Negative control 72 h	12.75	0.0039

## Data Availability

Data is contained within the article and Appendix A. Further inquiries can be directed to the corresponding authors

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
