# Peer review of "In Vitro Model of the Human Blood–Brain Barrier to Explore HTLV-1 Immunopathogenesis"

_cimb, 2025, doi:10.3390/cimb47100818_

Round 1
Reviewer 1 Report
Comments and Suggestions for Authors
NA

Author Response
1- You stated that MT2 was your positive control, but the cytokine release was higher in the sample. I am asking about the validity of positive control.
Dear reviewer, we deeply appreciate your constructive comments. To address this issue, we increased the number of positive controls to 4 (equalizing the number of cases). Regarding the use of immortalized cell lines, we agree that the use of immortalized cells presents phenotypic limitations compared to human primary cells. We have added a note to the text regarding the limitations of this model. Furthermore, we reinforce in the CONCLUSION the need for future validation in models with primary cell lines.
2- Your negative control was only one patient. Do you think the statistics would be better with 3 candidates?
Dear reviewer, we deeply appreciate your constructive comments. Regarding sample size and the number of negative controls, we recognize that the reduced number of negative control samples limits the statistical power of the study. To address this issue, we increased the number of negative controls to 4 (equalizing the number of cases).
3- You did not examine the BBB integrity with electrical stimulation. I do not know if the cytokine profile is enough to represent the integrity of the model.
Dear reviewer, we deeply appreciate your constructive comments. We added a new description in the METHODS section of a BBB integrity validation by the PatchClamp electrophysiology system.
4- In the sample processing, you said that PBMCs were centrifuged at 1500 RPM. I do not know how much G-force you used. I think it is a little bit fast to damage the cell or induce cytokine release as an artifact. please explain.
Dear reviewer, we deeply appreciate your constructive comments. We added a new description in the METHODS section.
5- Please indicate in the table that both cell lines' cytokine results are taken from the BBB model after co-culture. See Table 1 or Figure 1. Please add capitation with the number of the table.
Dear reviewer, we deeply appreciate your constructive comments. We added a new description in the RESULTS section.

Reviewer 2 Report
Comments and Suggestions for Authors
This study provides important new insights on the inflammatory processes behind HTLV-1-associated dementia. Still, other parts need improvement or explanation. The primary concern is the small sample size, particularly given the use of only one negative control, which significantly reduces the generalizability of the findings and statistical power. The debate acknowledges this, but the data interpretation throughout the paper underlines it only insufficiently. Another problem relates to the modeling of complex in vivo systems, including the blood-brain barrier and neurons, using immortalized cell lines (hBMEC and SH-SY5Y), without enough consideration of their physiological limits or phenotypic changes relative to primary human cells. The method falls short in defining the criteria for confluence and normalizing cytokine quantification data (e.g., per cell or protein content), as well as in handling batch effects or inter-donor variability in PBMCs. While IL-6 and IL-8 are important findings, there is no research showing how these cytokines directly cause the observed neuronal death; they are related, but we can't say one causes the other. Since earlier studies, including those mentioned in the paper, have already shown that IL-6/IL-8 develop and the blood-brain barrier is disrupted in HTLV-1 infection, the discussion makes the results seem more unique than they really are. Some statements in the discussion and conclusion suggest that treatments could focus on IL-6 or IL-8; however, there is no experimental evidence provided or talked about to support this idea.
Additionally, the ongoing mention of IL-6 and IL-8 patterns at different times and in various cell lines shows a lot of overlap that could be reduced without losing scientific understanding. Self-citations are generally acceptable, but they could bolster the central claims by incorporating more general literature. Although the scientific case is strong, the experimental technique and interpretation require improvement, with a greater focus on mechanical knowledge, biological relevance, and model limitations. Being more cautious and waiting for validation in larger, more physiologically relevant models would help improve the conclusions.
Final Advice: Conclusive and Important Amendment. Using a BBB in vitro model, the work investigates a relevant and poorly studied topic—the function of inflammatory cytokines in HTLV-1-associated myelopathy (HAM/TSP)—presenting intriguing preliminary evidence; yet it is hampered by major constraints that prevent its acceptance in its current state. The issues include having a small number of samples (especially just one negative control), not thoroughly exploring the biological limits of the in vitro model, being too different from previous studies, and mistakenly treating the correlation of cytokine data as a direct cause of neuronal death. The following significant changes are necessary to meet the publishing criteria: arguments supporting the physiological relevance of the experimental model. Additionally, a more cautious interpretation of the data is needed, particularly regarding the consequences of treatment. The discussion of cytokine data should be eliminated. We need to clarify the statistical methods and normalization guidelines. Extend the conversation to include relevant reading outside of personal references. Once we satisfactorily address these issues, we can approve the publication as a well-organized preliminary study with essential hypotheses for subsequent translational research.
Author Response
Dear reviewer, we deeply appreciate your constructive comments. We would like to clarify some points:
Regarding sample size and the number of negative controls, we recognize that the reduced number of negative control samples limits the statistical power of the study. To address this issue, we increased the number of positive controls to 4 (equalizing the number of cases).
Regarding the use of immortalized cell lines, we agree that the use of immortalized cells presents phenotypic limitations compared to human primary cells. We have added a note to the text regarding the limitations of this model. Furthermore, we reinforce in the CONCLUSION the need for future validation in models with primary cell lines.
Furthermore, we will revise the METHODS section to clarify the criteria used to determine cell confluence.
Furthermore, we recognize that suggesting treatments without direct experimental evidence may be premature. In this context, we have reformulated our conclusions, adopting a more cautious and speculative stance, making it clear that these are hypotheses that require future investigation.

Round 2
Reviewer 1 Report
Comments and Suggestions for Authors
NA
Author Response
We sincerely appreciate the time and effort devoted to reviewing our manuscript. Your insightful comments have been invaluable, helping us improve the clarity and quality of our work.

Reviewer 2 Report
Comments and Suggestions for Authors
I have reviewed the manuscript titled “In Vitro Model of a Human Blood-Brain Barrier to Explore the HTLV-1 Immunopathogenesis” and provide the following evaluation. The paper is well-motivated, addressing an important gap in understanding how HTLV-1 infection influences blood–brain barrier (BBB) integrity and contributes to neuroinflammation in HAM/TSP. The introduction clearly frames the clinical context and provides sufficient background, although it could be tightened to avoid repetition. The methodology is sound in terms of establishing an in vitro BBB model using hBMEC and SH-SY5Y co-cultures, with stimulation from PBMCs of different clinical groups and an MT-2 positive control. However, a significant limitation is the minimal sample size (n = 4 per group), which the authors acknowledge, but this reduces the robustness and generalizability of the findings. The TEER measurements, cytokine quantifications, and apoptosis assays are well described, but specific details, such as controls for technical variability, blinding, or replication across independent donors, could be more explicitly stated to strengthen the integrity of the work.
The results are clearly presented, with extensive data tables showing cytokine levels at multiple time points, and statistical analyses using Kruskal–Wallis followed by Fisher’s LSD post hoc tests with Bonferroni correction. The statistical approach is acceptable but not ideal for such small groups, and the interpretation occasionally overstates significance given the sample constraints. The discussion effectively situates the findings within the prior literature, demonstrating how elevated levels of IL-6, IL-8, IL-1β, TNF, and IL-12p70, along with increased IL-10, contribute to neuroinflammation and neuronal apoptosis. However, the discussion sometimes draws firm mechanistic conclusions without sufficient in vivo validation, and greater caution would improve the balance. The conclusions effectively emphasize the preliminary nature of the model; however, the manuscript could be more concise and avoid redundancy across sections.
With respect to citations, the reference list is extensive and generally appropriate, covering both classical and recent work. However, there is moderate reliance on a small set of authors and overlapping Fiocruz-associated publications, which raises the possibility of self-citation bias. Several references are recent and relevant, but in places the discussion leans heavily on older literature without integrating the most recent mechanistic or translational findings beyond HTLV-1 (e.g., HIV or multiple sclerosis BBB studies could be expanded for context).
In terms of areas for improvement, the main issues are the limited sample size, over-interpretation of preliminary findings, occasional redundancy in writing, and the need for a more critical discussion of limitations and broader translational implications. Figures could also be improved—particularly by integrating quantitative graphs alongside the extensive tables for better readability. A deeper consideration of whether cytokines identified could realistically serve as clinical biomarkers, or whether they are simply in vitro artifacts, would strengthen the translational impact.
In conclusion, I find the paper scientifically interesting and methodologically competent, but the limitations in sample size, analysis, and cautious interpretation must be addressed before it is suitable for publication. My recommendation is a significant revision, with requests for more precise articulation of limitations, improved presentation of data, tighter discussion, and more balanced integration of references.
Author Response
We sincerely appreciate the time and effort devoted to reviewing our manuscript. Your insightful comments have been invaluable, helping us improve the clarity and quality of our work. Below, we provide detailed responses and specify the revisions made to address your concerns.
About the sample size and robustness of the findings, we fully acknowledge the limitations posed by the small number of donors, as indicated in our manuscript. This was a deliberate choice reflecting the exploratory nature of this study, aimed at establishing a functional in vitro BBB model for HTLV-1 infection. We have now expanded the discussion section to more explicitly emphasize this limitation and caution interpretations accordingly. Additionally, we highlighted the need for validation with larger cohorts and more complex in vivo models to build on these preliminary results. Also, we agree with you that our data are preliminary. We believe that due to the scarcity of data in the literature, our results might encourage future studies to elaborate new experimental designs addressing the BBB involvement in HTLV infection. In addition to the limitations described in the discussion section, we continue to make an effort to increase the number of recruited participants for new future studies.
Concerning the controls and technical variability, we have added clarifications in the Materials and Methods section. We confidently emphasize that all PBMC samples utilized in this study were rigorously confirmed to be from patients who had undergone thorough testing for HTLV-1 infection. Considering the virus prevalence in the state of Pernambuco, according to the latest published epidemiological bulletin (https://www.gov.br/aids/pt-br/central-de-conteudo/publicacoes/2022/boletim_epidemiologico-svs-48-htlv.pdf), the document describes a rate of 2.2/1000 inhabitants positive for both types of HTLV (these numbers were obtained through blood testing, including screening and confirmation tests, which already represents a limited population). Altogether, the sample size (n=4) of HTLV infected patients might not be considered small. We agree with you that this is a preliminary study and it should be interpreted with caution. Even though it was not possible to implement full blinding throughout every experimental step due to practical constraints, data acquisition and subsequent analysis were performed in a semi-blinded manner. Specifically, the analysts were unaware of key sample group identities during critical stages of data processing. Moreover, software-assisted data analysis was employed to reduce subjective interpretation.
Concerning the data presentation and the figures, we improved data visualization by including a concise graphical representation of the apoptosis assays alongside extensive tabular data. This enhances the readability and interpretability of the results. We also streamlined the Results and Discussion to reduce redundancy and ensure a clearer, more focused narrative.
We carefully revised the Introduction to reduce repetitive statements and kept it concise while maintaining the scientific context and relevance. The Discussion now includes a more balanced view, with additional remarks about the limitations of our in vitro model — specifically that it cannot fully recapitulate the heterogeneity and dynamics of the human BBB in vivo. We stressed the preliminary character of the mechanistic interpretations and the need for additional investigations to substantiate these findings.
Addressing the potential self-citation bias, we reviewed our reference list and replaced some Fiocruz-related publications with other relevant and recent literature from the broader field, including studies on BBB involvement in other viral infections such as HIV and neurodegenerative diseases. However, we would like to clarify respectfully that, since our group specializes in this field and the suggested model is based on our own research, it makes sense that some of our earlier work would be referenced to set the scene and bolster the current investigation.
Finally, our discussion of cytokines as potential biomarkers was revised to explicitly note that these findings are based on an in vitro setting and require validation in clinical samples with longitudinal follow-up. We refrained from overinterpreting the current data and highlighted that while promising, clinical utility hinges on future confirmatory studies.

Round 3
Reviewer 2 Report
Comments and Suggestions for Authors
I have reviewed the revised manuscript titled “In Vitro Model of a Human Blood-Brain Barrier to Explore the HTLV-1 Immunopathogenesis”. The revisions have improved the clarity, methodological detail, and overall scientific rigor of the study. The manuscript now presents a coherent narrative, from introduction to conclusion, situating the work within the context of HTLV-1 neuropathogenesis and highlighting the novelty of using an in vitro BBB mimic model. The addition of detailed methodology (sample processing, co-culture conditions, TEER measurements, cytokine quantification, and apoptosis assays) strengthens reproducibility and transparency. The inclusion of comprehensive results—such as the cytokine quantifications in Table 1 (page 6) and neuronal apoptosis data in Figure 2 (page 11)—supports the stated conclusions that HTLV-1-infected PBMCs, particularly from HAM/TSP patients, trigger strong pro-inflammatory cytokine responses and neuronal cell death.
In terms of scientific integrity, the manuscript is ethically sound, with clear approvals, informed consent, and funding acknowledgments. The experimental design, though limited by a small sample size, is appropriate for a pilot study. The authors are transparent about limitations, explicitly noting the restricted cohort size and the simplified BBB model that excludes astrocytes, pericytes, and physiological flow. The discussion now compares findings with previous studies, reinforcing the significance of cytokines IL-6, IL-8, IL-1β, and TNF as potential biomarkers of neuroinflammation, while also noting IL-10’s dual anti-inflammatory role. This balanced interpretation reduces overstatement and acknowledges the need for in vivo validation.
There are still minor areas for improvement. Some results sections contain redundant phrasing, with repeated confirmation of statistical significance that could be streamlined for readability. The discussion could be sharpened by synthesizing findings more concisely, rather than restating individual cytokine roles multiple times. The figures and tables are informative but would benefit from clearer legends that fully explain abbreviations without requiring cross-reference to the text. While the citations are appropriate and up to date, a few earlier foundational references could be replaced with more recent meta-analyses to enhance currency.
Overall, the manuscript demonstrates significant improvement from prior versions, with strengthened methodological rigor, clearer results, and more cautious interpretation. Given its novelty and translational potential, I recommend acceptance with minor revision. The revisions should focus on reducing redundancy in the results and discussion, clarifying figure/table legends, and polishing references for consistency and currency.
Author Response
We sincerely appreciate your acknowledgment of the improvements in reproducibility and transparency achieved by including more detailed methods and expanded results. We are also grateful for your recognition of the study’s ethical rigor and the suitability of our experimental design for a pilot investigation. In response to your suggestions for minor revisions, we have addressed the following points:
- Reducing redundancy in the Results section: We have revised the text to eliminate repetitive statements regarding statistical significance, which has improved the clarity and flow of the presentation.
- Sharpening the Discussion: The discussion has been condensed to focus on a clear, integrated interpretation of the cytokine findings, avoiding repeated descriptions of individual cytokine roles.
- Clarifying figure and table legends: All abbreviations are now fully explained, and the legends have been expanded to ensure they are clear and understandable on their own, without needing to refer back to the main text.
- Updating references: We carefully reviewed the citations and determined that the foundational studies remain crucial for supporting the key points of our manuscript. Although replacing all earlier references with more recent meta-analyses was not feasible without losing important context, we have ensured that the reference list incorporates current literature where appropriate.
We believe these changes have further strengthened the manuscript. We thank you again for your valuable comments, which have greatly contributed to the quality of our work. Please find the revised manuscript along with a detailed point-by-point response to your feedback.